# The Many Faces of Post-Ischemic Tau Protein in Brain Neurodegeneration of the Alzheimer’s Disease Type

**DOI:** 10.3390/cells10092213

**Published:** 2021-08-27

**Authors:** Ryszard Pluta, Stanisław J. Czuczwar, Sławomir Januszewski, Mirosław Jabłoński

**Affiliations:** 1Laboratory of Ischemic and Neurodegenerative Brain Research, Mossakowski Medical Research Institute, Polish Academy of Sciences, 5 Str. Pawińskiego, 02-106 Warsaw, Poland; sjanuszewski@imdik.pan.pl; 2Department of Pathophysiology, Medical University of Lublin, 8b Str. Jaczewskiego, 20-090 Lublin, Poland; stanislaw.czuczwar@umlub.pl; 3Department of Rehabilitation and Orthopedics, Medical University of Lublin, 8 Str. Jaczewskiego, 20-090 Lublin, Poland; mbjablonski@gmail.com

**Keywords:** brain ischemia, hippocampus, tau protein, excitotoxicity, oxidative stress, apoptosis, autophagy, neuronal death, neuroinflammation, mitochondrial dysfunction, neurofibrillary tangles, dementia, neurodegeneration, gen

## Abstract

Recent data suggest that post-ischemic brain neurodegeneration in humans and animals is associated with the modified tau protein in a manner typical of Alzheimer’s disease neuropathology. Pathological changes in the tau protein, at the gene and protein level due to cerebral ischemia, can lead to the development of Alzheimer’s disease-type neuropathology and dementia. Some studies have shown increased tau protein staining and gene expression in neurons following ischemia-reperfusion brain injury. Recent studies have found the tau protein to be associated with oxidative stress, apoptosis, autophagy, excitotoxicity, neuroinflammation, blood-brain barrier permeability, mitochondrial dysfunction, and impaired neuronal function. In this review, we discuss the interrelationship of these phenomena with post-ischemic changes in the tau protein in the brain. The tau protein may be at the intersection of many pathological mechanisms due to severe neuropathological changes in the brain following ischemia. The data indicate that an episode of cerebral ischemia activates the damage and death of neurons in the hippocampus in a tau protein-dependent manner, thus determining a novel and important mechanism for the survival and/or death of neuronal cells following ischemia. In this review, we update our understanding of proteomic and genomic changes in the tau protein in post-ischemic brain injury and present the relationship between the modified tau protein and post-ischemic neuropathology and present a positive correlation between the modified tau protein and a post-ischemic neuropathology that has characteristics of Alzheimer’s disease-type neurodegeneration.

## 1. Introduction

A dangerous consequence of an ischemic episode to the brain is the massive death of neurons with the progressive occurrence of neurodegeneration and the development of finally full-blown dementia [1,2,3,4,5,6]. The phenomenon of post-ischemic brain neurodegeneration is considered to be the most common cause of late-onset dementia in the world. The prevalence of dementia after a primary and recurrent stroke is estimated at 10 and 41%, respectively [7]. Worldwide, post-ischemic dementia occurs in ~50% of cases, depending on the diagnostic criteria and geographic location [4]. In fact, it has already been proven that post-ischemic dementia shares many features with dementia in Alzheimer’s disease [8]. As people age, the number of cases with dementia is projected to reach 82 million by 2030 and 152 million by 2050 [9]. Overall, the number of dementia patients is increasing rapidly with the aging of the human population, reflecting the urgent health situation worldwide, with pathological processes not fully understood and with no actually known treatments.

The development of post-ischemic dementia is also observed in animals [10,11,12,13,14]. Dementia in animals and humans positively correlates with an increased number of damaged and dying neurons and with progressive neuroinflammatory changes, especially in the hippocampus [11,15,16,17]. The dementia phenomenon develops steadily and slowly with an increasing survival time after ischemia [11]. The development of dementia is well correlated with overall brain atrophy [1,2,18,19,20,21,22]. Dementia following an ischemic brain injury is irreversible and lasting [11].

The mechanisms of post-ischemic brain neurodegeneration with the development of full-blown dementia are complex and unclear, influenced by multiple mediators. It has been suggested that changes in calcium levels, hyperactivity of the glutamate system, acetylcholine deficiency, and metal ion dyshomeostasis are closely related to post-ischemic neurodegenerative pathways [23,24,25,26]. In addition, processes such as oxidative stress, apoptosis, neuroinflammatory changes, and impaired autophagy have been shown to cause severe brain damage and contribute to chronic and irreversible changes following transient focal or global ischemia [16,17,27,28,29,30,31,32]. The excessive generation of amyloid and the increased dysfunction of the tau protein are currently the most studied elements in the ischemic neurodegeneration of the brain with the development of full-blown dementia [33,34,35,36]. An increased level of amyloid in the brain after ischemia has been reported and it causes the development of amyloid plaques and cerebral amyloid angiopathy [21,37]. A reduction in cerebral blood flow in the brain after ischemia due to vasoconstriction [38] and the advancement of cerebral amyloid angiopathy [21,37] inhibits the transport of nutrients to the brain, and also reduces the removal of the neurotoxic amyloid and/or tau protein by the ischemic blood-brain barrier [37].

It has been suggested that ischemic brain injury in humans and animals is associated with the successive incidence of Alzheimer’s disease-type neuropathology [35]. In this situation, an understanding of the basic slow pathological mechanisms in connection with ischemic progressive brain injury is now required. In this review, we discuss the post-ischemic mechanisms related to the genotype and phenotype of Alzheimer’s disease, which interconnect the emergence of the increased expression of the tau protein gene and its modified products. This is due to the emergence of a large amount of new evidence for genomic and proteomic changes in the tau protein in humans and animals after cerebral ischemia. The disclosure of increased post-ischemic tau protein gene expression sheds new light on a better understanding of the modified tau protein as the cause of the effects of brain ischemia in clinical and animal studies. Even though significant advances in the post-ischemic pathology of tau protein studies have recently been made, the processes underlying tau protein-induced neurodegeneration following ischemia are still unclear. Below, we present an outline of the relationship of the tau protein with the neurodegenerative mechanisms following cerebral ischemia typical of Alzheimer’s disease. The current review aims to update the facts regarding the relationship between post-ischemic brain disease and the promotion of Alzheimer’s disease-type neuropathology. As a result, a final understanding of the neural pathways correlated with ischemic injury and death, and the discovery of possible new pathogenic mechanisms in post-ischemic stroke is important for the successful treatment of post-ischemic stroke sequalae. Such studies can help determine the need for innovative treatments for ischemic stroke in the clinic and may be important in organizing and assessing potential priorities for prevention. New evidence clearly suggests an association between post-ischemic dementia and the ischemic neuropathological changes in the amyloid and the tau protein that are characteristic of Alzheimer’s disease. In this review, we first focus on assessing the response of the tau protein gene and its products to a reversible episode of brain ischemia. Secondly, we will present the role of the tau protein after brain ischemia in the development of neuropathological changes characteristic of Alzheimer’s disease, focusing on changes in its structure in the post-ischemic period. In this review, we will also look at possible future treatment strategies to slow or reduce neuropathological responses following brain ischemia associated with tau protein modifications.

## 2. Post-Ischemic *Tau Protein* Gene Expression in the Brain

Recently, only two experimental reports show an association between hippocampal regions CA1 and CA3 and alterations in the post-ischemic expression of the *MAPT* gene within 2, 7, and 30 days [33]. In the CA1 area, the expression of the *MAPT* gen considerably increased on the second day after ischemia (Table 1) [33]. Conversely, the expression of the *MAPT* gene decreased on days 7 and 30 (Table 1) [33].

In the post-ischemia area of CA3, a decreased expression of the *tau protein* gene was observed on day two (Table 1) [36]. In contrast, the expression of the *tau protein* gene increased 7–30 days after ischemia (Table 1) [36].

## 3. Post-Ischemic Tau Protein Accumulation in the Brain

Historical studies have shown a strong accumulation of the tau protein in neurons, astrocytes, and oligodendrocytes in the hippocampus, thalamus, and cortex in both experimental [39,40,41,42,43,44] and post-ischemic brain injuries in humans [45,46,47,48,49]. The tau protein was also accumulated in microglia in ischemic penumbra [47,48,49]. The above observations indicate that some neurons show changes in the tau protein after brain ischemia with reperfusion [41], which may indicate the main pathological phase of the development of ischemic processes in these cells [43]. Another study showed that the altered tau protein blocks the movement of organelles, neurofilaments, amyloid protein precursor vesicles, and increases oxidative stress in the neuronal body, axons, and dendrites, leading to the accumulation of the amyloid protein precursor in neuronal cells [50]. In addition, the levels of total tau protein, which were tested using brain microdialysis, increased in the brain following ischemia induced by cardiac arrest [51].

## 4. Post-Ischemic Tau Protein in the Blood after Brain Injury

An increase in the tau protein was found in the blood following global ischemic brain injury due to cardiac arrest with two peaks on survival days two and four, indicating progressive neuronal damage during recirculation [52]. The observed two-stage increase in the blood tau protein concentration is consistent with the following two types of neuronal death: the first is as a result of necrosis and the second is as a result of programmed neuronal death [53]. The profiles appear likely to reflect the time course of acute and delayed ischemic neuronal damage/death due to cardiac arrest [53]. These observations suggest that the blood tau protein levels may be a prognostic indicator of neurological recovery in post-ischemic brain injury due to cardiac arrest [52,53].

The increase in tau protein has also been documented in blood samples after ischemic stroke in humans, and for the most part, is a likely sign of damage progression to neuronal bodies and their axons following ischemia [54,55,56,57,58]. An increase in tau protein levels has also been reported in the cerebrospinal fluid of patients after an acute ischemic stroke [58]. Matrix metalloproteinase-9 has been reported to contribute to an increase in the blood tau protein in humans in the course of an acute ischemic stroke [55]. The increase in plasma tau protein pessimistically correlates with clinical outcomes following cerebral ischemia insult.

## 5. Post-Ischemic Tau Protein Hyperphosphorylation in the Brain

Following completely reversible cerebral ischemia insult due to cardiac arrest, the tau protein was re-phosphorylated and accumulated [59]. However, a transient local ischemic brain injury in a one-day surviving rat causes specific hyperphosphorylation of the tau protein in the vicinity of the injury (Table 2) [60]. In the case of the death of pyramidal neurons in the CA1 region of the hippocampus after forebrain ischemia in a gerbil, the hyperphosphorylation (Table 2) at serine 199/202 of the tau protein is controlled by mitogen activated protein kinase, cyclin dependent kinase 5 and glycogen synthase kinase 3 [61]. Recent studies indicate that after ischemia, the tau protein is hyperphosphorylated (Table 2) in cortical neurons and is associated with the development of apoptosis [48,49,60,62,63,64]. These data support the claim that post-ischemic neuronal apoptosis is closely related to the hyperphosphorylation of the tau protein. In addition, ischemic brain injury with hyperhomocysteinemia leads to ~700 times more hyperphosphorylated tau protein-positive neurons in the hippocampus and cortex compared to the control animals [65].

## 6. Post-Ischemic Tau Protein and Neurofibrillary Tangle Development in the Brain

The above facts indicate that the tau protein is highly hyperphosphorylated after reversible brain ischemia (Table 2). This triggers the formation of paired helical filaments after cerebral ischemia [68], neurofibrillary tangle-like [60,63] and neurofibrillary tangles [70,71] typical of Alzheimer’s disease (Table 2) (Figure 1). The development of neurofibrillary tangles was observed in the Meynert basal nucleus on the same side as human cerebral infarction [70,71]. In support of the above fact, elevated levels of cyclin-dependent kinase 5, involved in the development of neurofibrillary tangles (Table 2) (Figure 1), have been reported following experimental ischemic brain injury [63]. This may indicate an involvement of the modified tau protein in the death of neurons in the hippocampus post-ischemia (Figure 1). The above evidence also indicates/explains the different regulation during the ischemic death of pyramidal neurons in the CA1 and CA3 areas of the hippocampus in a manner dependent on the amount and changes in the structure of the tau protein (Table 1).

## 7. Post-Ischemia Tau Protein Intersection with Multiple Overlapping Phenomena/Pathologies in Brain Neurodegeneration

The functions of the tau protein are controlled by a multifaceted system of post-translational changes such as glycation, phosphorylation, acetylation, nitration, isomerization, O-GlcNAcylation, sumoylation, and truncation [72,73,74], suggesting that the tau protein plays an important role in both the physiology and pathology of the brain [75]. The modified structure of the tau protein is one of the most neurotoxic proteins accumulated in neuronal and neuroglial cells post-ischemia in humans and animals [27,28,33,34,36,47,63,69,70]. According to previous studies, the stages of the dysfunctional tau protein differ in different ischemic brain models such as dephosphorylation [32,41,42,64,66,76,77], re-phosphorylation [32,66], hyperphosphorylation [32,49,60,62,63], and the development of neurofibrillary tangles [69,70,71] (Table 2) (Figure 1). The hyperphosphorylated form of the tau protein in the brain post-ischemia reduces the affinity of the tau protein for microtubules by disrupting the binding balance [32,49,60,62,63,65,78]. In the following part of the review, we will discuss the main points of possible pathological tau protein activity in post-ischemic brain neurodegeneration.

### 7.1. Post-Ischemic Tau Protein versus Blood-Brain Barrier

Hyperphosphorylation of the tau protein after ischemic brain injury [48,49,60,62,63,64,65,70,71,79,80] triggers the development of neurofibrillary tangles [63,70,71], which are one of the major components of pathology in the brains of Alzheimer’s disease patients. An ischemic brain injury causes a pathological permeability of the blood-rain barrier [81,82,83,84,85], which also affects the hyperphosphorylation of the tau protein [48,49,60,63,64,65,67,68,70,71,79,80], and the modified tau protein may cause an additional exacerbation of blood-brain barrier dysfunction (Figure 2), which induces harmful feedback [86]. An accumulation of amyloid in the brain, associated with the ischemic permeability of the blood-brain barrier [87,88], may, in a roundabout manner, allow the onset of tau protein dysfunction, supporting the automatic link between amyloid accumulation and tau protein modification at some stage of blood-brain barrier breakdown [86]. Moreover, both oxidative stress [89] and neuroinflammation [16,17] cause damage to the blood-brain barrier that may cause hyperphosphorylation of the tau protein and the development of neurofibrillary tangles post-ischemia [63,70,71,90]. Moreover, after ischemia, the plasma-derived tau protein [52,53] crosses the ischemic blood-brain barrier in two directions and can enhance its own pathology in the brain [91]. In summary, ischemic blood-brain barrier failure may exacerbate in the brain tau protein neuropathology in post-ischemic brain injury and also suggests that ischemic brain pathology may be part of the cause responsible for the increase in the serum tau protein concentration [52,53,91,92].

### 7.2. Post-Ischemic Tau Protein versus Excitotoxicity

Excitotoxicity has been identified as one of the most important pathological mechanisms associated with calcium changes in post-ischemic brain injury [23,93,94]. The existing data suggest that tau protein phosphorylation can be inhibited by reducing calcium influx into neurons [95]. It has been revealed that impaired glutamate homeostasis or the elevated activity of calcium-dependent kinases may induce tau protein phosphorylation [96,97], and consequently, glutamate-induced cytotoxicity may exacerbate the dysfunctional appearance of the tau protein (Figure 2) [74]. Conversely, many studies have shown that the tau protein also plays a significant role in enhancing excitotoxicity [58,98,99,100,101,102]. In P301L tau protein mice, KCl evoked an increase in glutamate release and decreased glutamate clearance in the hippocampus [102]. The exact mechanisms underlying tau protein-induced excitotoxicity require further elucidation. One study shows that the tau protein increases excitotoxicity without increasing calcium influx through the kainic acid receptor [103]. On the other hand, other studies suggest that reducing tau protein phosphorylation at Y18 may reduce N-methyl-d-aspartic acid receptor-mediated excitotoxicity in neurons [104,105]. Overall, the phenomenon of excitotoxicity with the phosphorylation of the tau protein leads to a vicious circle with respect to neuronal death in post-ischemic neurodegeneration (Figure 2).

### 7.3. Post-Ischemic Tau Protein versus Oxidative Stress

Oxidative stress is involved in neuropathological processes in the brain after ischemia in animals and humans. In experimental models of ischemic neurodegeneration, it has been established that the hyperphosphorylation of the tau protein may be a product of oxidative stress (Figure 2) [74,106,107]. Thus, tau protein hyperphosphorylation might be reduced using antioxidants [74,108,109,110]. There is no definite opinion about the causal interaction between oxidative stress and tau protein hyperphosphorylation. Some studies have shown that products of thiobarbituric acid, polyunsaturated lipids, and 4-hydroxynonenal, resulting from cell lipid peroxidation, are significantly increased, which can cause tau protein hyperphosphorylation [74,106,109]. Recently, it has been suggested that the hyperphosphorylation of the tau protein is due to the direct influence of reactive oxygen species, which is generated by 1,2-diacetylbenzene as a result of the phosphorylation of activated glycogen synthase kinase 3β [74,107]. Moreover, high levels of the hyperphosphorylated tau protein have been documented to initiate the production of reactive oxygen species (Figure 2). Ultimately, oxidative stress and the hyperphosphorylated tau protein may be two critical elements of the vicious cycle in the development of post-ischemic brain neurodegeneration (Figure 2).

### 7.4. Post-Ischemic Tau Protein versus Mitochondria

The activity of neurons is closely related to energy deficiency. Thus, the task of the mitochondria is to continually supply energy to neuronal and neuroglial cells. Consequently, impaired mitochondrial activity is an important neuropathological process in the brain following ischemia with subsequent recirculation. Dysfunctional mitochondrial activity is closely related to neuronal autophagy, necrosis, and apoptosis [111]. Mitochondrial stability conditioned by fusion and fission is a major issue in the development of mitochondrial dysfunction. Earlier data showed that protein 1 is related to dynamin, a mitochondrial fission protein, and may work together with the phosphorylated tau protein to induce mitochondrial dysfunction (Figure 2) [112,113]. A reduction in dynamin-related protein 1 protects against the hyperphosphorylated tau protein-induced dysfunction of mitochondria [114]. In a murine model of tauopathy, tau protein deposits undermine the distribution of mitochondria in neuronal cells [115]. The unusual behavior of mitochondria can be improved by reducing the level of soluble tau protein in their environment [51,115]. Tau protein accumulation can both damage normal activity and mitochondrial allocation by increasing mitofusins, which can cause ATP depletion, the development of oxidative stress, and synaptic abnormalities [116,117,118]. The pathway studies used axonal protein phosphatase 1, glycogen synthase kinase 3, and the retention of the C-Jun amino-terminal kinase-interacting protein 1 kinesin motor protein complex by phosphorylated tau protein, which may be involved in neuropathological interactions [119,120]. It should also be noted that tau protein phosphorylation can also be enhanced by reactive oxygen species, mimicking mitochondrial oxidative stress in neurons [121]. In summary, the dysfunction of the tau protein may disrupt the function and dynamics of mitochondria, and such altered mitochondria may be an indicator of tau protein phosphorylation and aggregation (Figure 2).

### 7.5. Post-Ischemic Tau Protein versus Autophagy

It is well known that autophagy plays a key role in the maintenance of normal levels of tau protein in neuronal cells [122,123,124]. Autophagy has been shown to be an important neuropathophysiological process in brain neurodegeneration after an ischemic stroke [125]. Previous research has shown that a decrease in the tau protein is correlated with an increase in an autophagy marker such as microtubule-associated protein 1A/1B-light chain 3B-II in a 3xTg mouse model of Alzheimer’s disease after reversible hypoperfusion, indicating that autophagy may be a way to reduce the dysfunctional tau protein levels in the brain [126]. In contrast, another study reported a significant reduction in microtubule-associated protein 1A/1B-light chain 3B protein growth and a reduction in infarct size in the P301L-Tau mouse model after ischemia [127]. It might be probable that autophagy insufficiency is triggered by a mutant tau protein with increased levels of its aggregates [127]. In addition, it has been documented that autophagy can induce tau protein expression in neuronal cells that overexpress the human P301L-Tau mutant [128]. In human tauopathies, p62 is an autophagy regulatory protein and its immunostaining co-localizes with tau protein inclusions [129]. In transgenic mice, the activity of autophagy may increase the clearance of the tau protein [130] and thus, reduce the aggregation of the seeded tau protein [131]. The phosphorylation of the tau protein is believed to be due to seeded aggregation [132]. The P62 and nuclear dot 52 protein are among the autophagy cargo receptors playing an important role in protecting against the aggregation of the seeded tau protein in neurons [128,133]. It is, therefore, highly likely that autophagy, not proteasomes, reduces the aggregation of the seeded tau protein (Figure 2) [128].

### 7.6. Post-Ischemic Tau Protein versus Apoptosis

Apoptosis is naturally programmed cell death, acting as the most important and dangerous neuronal killer following brain ischemia [134]. Tau protein hyperphosphorylation and apoptosis are believed to be two self-contained, self-sufficient, and overlapping neuropathological processes during neuronal death (Figure 2), although most researchers have found no significant relationship between these phenomena [135,136]. However, some studies have shown an ischemic accumulation of cyclin-dependent kinase-5 [63], which regulates tau protein phosphorylation, and may initiate neuronal apoptosis through degradation of the endoplasmic reticulum [137]. It has also been documented that hyperphosphorylation of the tau protein can be prevented by knocking down cyclin-dependent kinase-5, which may protect neuronal cells by alleviating endoplasmic reticulum stress from apoptosis [137]. Recent studies indicate that after cerebral ischemia, hyperphosphorylated tau protein accumulates in cortical neurons and is associated with their apoptosis (Figure 2) [48,49,60,62,63,64]. The above data clearly indicate that neuronal apoptosis after cerebral ischemia is associated with the hyperphosphorylation of the tau protein (Figure 2).

### 7.7. Post-Ischemic Tau Protein versus Neuroinflammation

Neuroinflammation is considered a pathway that influences neuronal death in the acute and chronic phase following cerebral ischemia with reperfusion [116]. Some previous studies have suggested that the dysfunctional tau protein is directly related to the neuroinflammatory cascade (Figure 2). It should also be noted that neuroinflammatory mediators can significantly affect the function and structure of the tau protein post-ischemia [73,138,139]. In addition, it has been suggested that the dysfunctional tau protein may be a trigger of the neuroinflammatory cascade (Figure 2) [73,138,139]. The exact role of neuroinflammatory processes in the post-ischemic neuropathology of the tau protein or the dysfunctional tau protein in neuroinflammation still needs to be clarified. Some researchers consider neuroinflammation as a worsening factor [137], but another study has found that neuroinflammation can lower the level of oligomeric tau protein by improving phagocytosis via microglia [140]. The first direct evidence for the involvement of neuroinflammation in tau protein pathology was presented in an in vitro study and showed that neuroinflammatory mediators, i.e., interleukin-1β, can promote tau protein hyperphosphorylation (Figure 2) by the stimulation of p38 mitogen-activated protein kinases [141]. This was also confirmed in the 3xTg model of Alzheimer’s disease in vivo with the development of plaques and tangles [142]. Recent studies have also shown that various stressors such as lipopolysaccharide, infection, and tumor necrosis factor-α can initiate an exacerbation of tau protein hyperphosphorylation [143,144,145]. As a consequence, lowering tau protein levels or inhibiting neuroinflammatory mediators may act as a treatment for tauopathies [146]. A study by Kovac’s group revealed a new toxic form of the misfolded tau protein, i.e., the formation of a truncated tau protein [72]. The truncated tau protein may increase the permeability of the blood-brain barrier (Figure 2) [72]. In addition, studies have also provided evidence that the truncated tau protein had a cytotoxic effect on astrocyte-microglia culture as manifested by increased levels of extracellular adenylate kinase. The blood-brain barrier damage induced by the truncated tau protein was mediated by the pro-inflammatory cytokine tumor necrosis factor α and the chemokine monocyte chemotactic protein 1 [72]. It should also be noted that the pro-inflammatory cytokine interferon-γ has been found to have an opposite effect on tau protein phosphorylation and dephosphorylation, and, ultimately, induced neurogenesis [147]. Microglial cells and macrophages play a very important role in neuroinflammation. Extracellular tau protein oligomers can be moderately phagocytosed by both microglia and macrophages under normal conditions [140]. Microglial internalization has been shown to be effective for both aggregated and soluble tau protein in vitro and in vivo [148]. Overall, the inhibition of neuroinflammation in the parenchyma of the brain may paradoxically be involved in the development of the neuropathology of the tau protein. In assessing the above information, more research is needed to elucidate these molecular phenomena.

## 8. Conclusions

In this review, we described the neuropathological role of the tau protein after brain ischemia. The presented data demonstrate that the tau protein plays a very important role, not only in the stabilization and structure of microtubules, but also in the neuropathology of the brain after ischemia (Figure 1). We then presented the various pathological stages of the tau protein following an ischemic brain injury. We further demonstrated the influence of the tau protein and its potential pathological forms on post-ischemic brain tissue structure through the effects on apoptosis, oxidative stress, excitotoxicity, autophagy, neuroinflammation, changes in the blood-brain barrier, and mitochondrial dysfunction. The hyperphosphorylation of the tau protein is the most important neuropathological phenomenon associated with the tau protein in an ischemic brain (Figure 1), (Table 2). For this reason, keeping tau protein phosphorylation under control may have potential post-ischemic protective effects. Some studies have found that the regional relocation of phosphorylated tau protein in an ischemic brain with reperfusion is similar to the changes in Alzheimer’s disease (Figure 1) [49]. Animal studies show an important role for the tau protein in a post-ischemic brain injury, suggesting that substances that target different forms of the tau protein have a great potential to reduce the consequences of brain neurodegeneration following cerebral ischemia. Clinical studies have shown that the level of the tau protein in the blood or cerebrospinal fluid is directly related to the severity of an ischemic stroke and its long-term sequelae. In summary, we presented the possible mechanisms/effects of post-ischemic tau protein in the brain, including excitotoxicity, oxidative stress, autophagy, apoptosis, neuroinflammation, changes in the blood-brain barrier, dysfunction of mitochondria, and finally, the development of neurofibrillary tangles (Figure 1). The above observations show that the tau protein may be at the intersection of many pathological mechanisms/processes/phenomena leading to severe neurodegenerative changes in the brain after ischemia. The facts show that an ischemic brain injury induces neuronal damage and death in the hippocampus via a tau protein-dependent mechanism, defining a new phenomenon that influences the long-term survival and/or death of post-ischemic neurons (Table 2, Figure 1) [33,34,41,42,49,60,63,64,67,68,149,150,151]. The underlying processes of neuronal death following ischemia include neuropathological effects in the tau protein. There are insufficient clinical studies focused on the relationship between the modified tau protein and the consequences of an ischemic stroke. We believe that the final disclosure of the pathological effects of the tau protein in an ischemic brain and post-ischemia and the influence of ischemia on tau protein phosphorylation may lead to the development of a possible innovative target for post-ischemic brain treatment.

## Figures and Tables

**Figure 1 cells-10-02213-f001:**
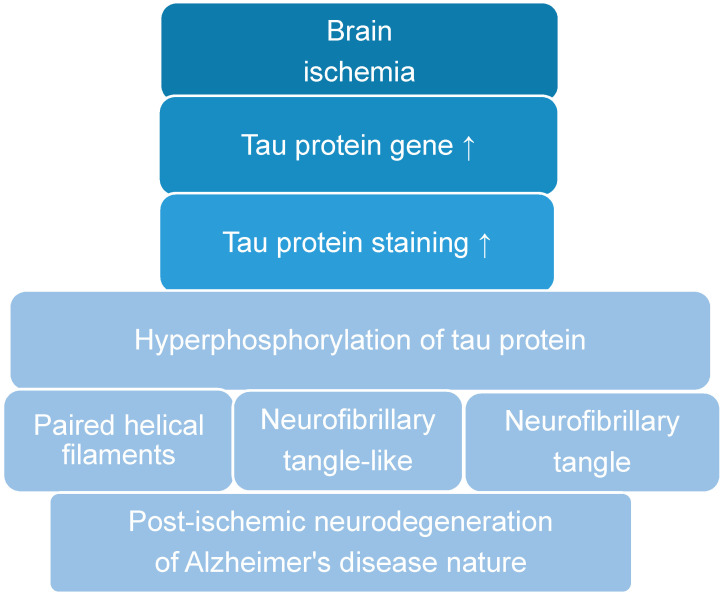
The role of post-ischemic tau protein in brain neurodegeneration. ↑—increase.

**Figure 2 cells-10-02213-f002:**
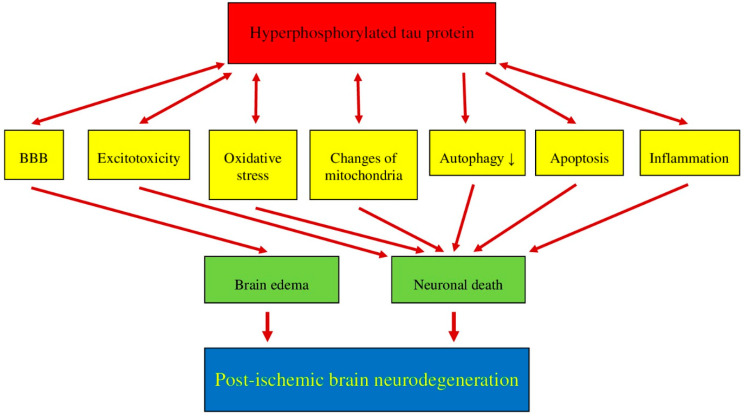
Interrelationships between hyperphosphorylated tau protein and post-ischemic brain neurodegeneration. ↓—decrease. BBB—blood-brain barrier.

**Table 1 cells-10-02213-t001:** Post-ischemic *tau protein* gene expression in the CA1 and CA3 area of the hippocampus on different days of survival.

	Days	2	7	30
Area				
CA1		↑↑↑	↓	↓
CA3		↓	↑	↑

Increase—↑, decrease—↓.

**Table 2 cells-10-02213-t002:** Hyperphosphorylation and the structure of the tau protein after brain ischemia.

Tau Protein	Ischemia	Human/Animal	Sites of Phosphorylation	Reference
Hyperphosphorylation	Focal	Rat	Asp421, pT181, pT205 pT212, pT231, pS202 pS214, pS262, pS396, pS404, pS422	[49,60]
Hyperphosphorylation	Global	Rat	Ser202, Ser262, Ser396 Thr205	[48,65]
Hyperphosphorylation	Forebrain	Gerbil	Ser199, Ser202	[61,66]
Hyperphosphorylation	Focal	Mouse	Ser262, Ser 356	[67]
Hyperphosphorylation	Stroke	Human	Ser101	[47]
Paired helical filaments	Forebrain	Mouse	pS396, pS404	[68]
Fibrillar tau protein	Focal + amyloid	Rat	Tau 2	[69]
Neurofibrillary tangle-like	Focal	Rat	P-396, P-404	[63]
Neurofibrillary tangles	Stroke	Human	Tau 1	[70,71]

## Data Availability

Not applicable.

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
