# Peer review of "The Many Faces of Post-Ischemic Tau Protein in Brain Neurodegeneration of the Alzheimer’s Disease Type"

_cells, 2021, doi:10.3390/cells10092213_

Round 1

Reviewer 1 Report

It is well known that Tau protein plays a key role in neuronal damage and clinical pathophysiology in several human diseases including post ischemic stroke. Among the multiple mediators involved, it seems that tau protein is at the intersection of many pathological mechanisms related with post-ischemic progressive brain injury. In the current review the authors try to summarize all these interrelationships regarding tau protein and postischemic brain changes. The review is interesting and well organized. Although the data are clearly presented the abundant information on the factors interacting with/affecting  tau protein hyperphosphorylation are somehow confusing for the non expert. Thus, a scheme, in addition to figure 1 would be of great help (i.e tau protein, autophagy and and LC3).

Minor comments

-Define several terms in the text when they appear for the first time or make a glossary (i.e  LC3: microtubule-associated protein 1A/1B-light chain 3;  GSK3β: Glycogen synthase kinase 3 beta; etc)

- minor editing is needed

Author Response

It is well known that Tau protein plays a key role in neuronal damage and clinical pathophysiology in several human diseases including post ischemic stroke. Among the multiple mediators involved, it seems that tau protein is at the intersection of many pathological mechanisms related with post-ischemic progressive brain injury. In the current review the authors try to summarize all these interrelationships regarding tau protein and postischemic brain changes. The review is interesting and well organized. Although the data are clearly presented the abundant information on the factors interacting with/affecting tau protein hyperphosphorylation are somehow confusing for the non expert. Thus, a scheme, in addition to figure 1 would be of great help (i.e tau protein, autophagy and and LC3).

All changes are marked in red. Thanks for your useful comments.

As suggested by the Reviewer, we made Figure 2.

Minor comments

-Define several terms in the text when they appear for the first time or make a glossary (i.e LC3: microtubule-associated protein 1A/1B-light chain 3; GSK3β: Glycogen synthase kinase 3 beta; etc)

In the text, we have presented a dozen or so terms without using any abbreviations.

- minor editing is needed.

Done.

Reviewer 2 Report

The review is devoted to relevant and rapidly developing theme. In this work the authors review the post-ischemic tau protein hyperphosphorylation in brain and its influence on a neurodegenetarion processes. They analyzed the role of post-ischemic tau protein at many aspects concomitant neurodegeneration such as apoptosis, oxidative stress, excitotoxicity, autophagy, neuroinflammation, changes in the blood-brain barrier, and mitochondrial dysfunction. However, there are many significant issues that need to be eliminated to publication.

 Major Points:

1. Section 2. Post-ischemic tau protein gene modification in brain. Line 102. The information provided is identical to the information in the article https://doi.org/10.3390/ijms21030892. It is required to rewrite this section, adding new information.

  1. Section 3. Post-ischemic tau protein staining in brain. Line 117. It is better not to use the term staining, but replace it with accumulation. If you want to write about staining, you should indicate what methods and dyes reveal the tau proteins and its aggregates.
  2. Section 5. Post-ischemic tau protein hyperphosphorylation in brain. Line 152-155. The authors discuss about hyperphosphorylation of the tau protein at serine 199/202 in gerbil and which kinases control this process. Add about other organisms. Table 2. Line 162.  It should be completed about the sites of phosphorylation of the tau protein in the table.

4. Subsection 7.1. Post-ischemic tau protein versus blood-brain barrier. Line 205. The information provided is similar to the information in the article https://doi.org/10.3390/ijms21030892. Remove the Subsection 7.1. or significantly rewrite adding new information.

  1. The authors review a rapidly developing theme, but there are very few new references. Updating the references, for example 18, 38-46, 96, will improve the work.

Minor points:

  1. Line 102. Post-ischemic tau protein gene modification in brain. The term modification in the Section 2 is not applicable, since the text talks about changes in expression of the tau protein gene, and not about its modification. Please replace with a more appropriate term, e.g. expression.
  2. Line 254. Please decrypt DAB in the text.
  3. Line 254. Please decrypt ROS in the text.

Author Response

The review is devoted to relevant and rapidly developing theme. In this work the authors review the post-ischemic tau protein hyperphosphorylation in brain and its influence on a neurodegenetarion processes. They analyzed the role of post-ischemic tau protein at many aspects concomitant neurodegeneration such as apoptosis, oxidative stress, excitotoxicity, autophagy, neuroinflammation, changes in the blood-brain barrier, and mitochondrial dysfunction. However, there are many significant issues that need to be eliminated to publication.

All changes are marked in red. Thanks for your useful comments.

Major Points:

  1. Section 2. Post-ischemic tau protein gene modification in brain. Line 102. The information provided is identical to the information in the article https://doi.org/10.3390/ijms21030892.It is required to rewrite this section, adding new information.

We removed two sentences that were identical to the cited work. We regret to say that the rest of the text is not identical to the quoted work, the data is presented in a different way. We must also add that there is no other data on this subject in the world literature.

2. Section 3. Post-ischemic tau protein staining in brain. Line 117. It is better not to use the term staining, but replace it with accumulation. If you want to write about staining, you should indicate what methods and dyes reveal the tau proteins and its aggregates.

We changed staining to accumulation as suggested by the Reviewer.

3. Section 5. Post-ischemic tau protein hyperphosphorylation in brain. Line 152-155. The authors discuss about hyperphosphorylation of the tau protein at serine 199/202 in gerbil and which kinases control this process. Add about other organisms. Table 2. Line 162. It should be completed about the sites of phosphorylation of the tau protein in the table.

In line 152-155 we have given the names of the kinases that control tau protein phosphorylation.

Now Table 2 lists the tau protein phosphorylation sites.

4. Subsection 7.1. Post-ischemic tau protein versus blood-brain barrier. Line 205. The information provided is similar to the information in the article https://doi.org/10.3390/ijms21030892. Remove the Subsection 7.1. or significantly rewrite adding new information.

We must also add that there is no other data on this subject in the world literature. Sorry, the way of writing the text does not coincide with the text quoted by the reviewer. We cannot delete this part of the work as it will spoil the entire layout of the work and that part of the work which is the theme of the article will be lost.

5. The authors review a rapidly developing theme, but there are very few new references. Updating the references, for example 18, 38-46, 96, will improve the work.

The presented works are important and there are no other studies on this subject in the literature apart from the added items 1 and 2 of the literature.

Minor points:

1. Line 102. Post-ischemic tau protein gene modification in brain. The term modification in the Section 2 is not applicable, since the text talks about changes in expression of the tau protein gene, and not about its modification. Please replace with a more appropriate term, e.g. expression.

 Done.

2. Line 254. Please decrypt DAB in the text.

Done.

3. Line 254. Please decrypt ROS in the text.

Done.

Round 2

Reviewer 2 Report

The comments have been fully eliminated, there are no new comments to the submitted work.

Author Response

We have done the editor's recommendations. Thanks for improving recommendations.